# Investigation of Loss of Shape Stability in Textile Laminates Using the Buckling Method

**DOI:** 10.3390/polym14214527

**Published:** 2022-10-26

**Authors:** Ludmila Fridrichová

**Affiliations:** Technical University of Liberec, 461 17 Liberec, Czech Republic; ludmila.fridrichova@tul.cz

**Keywords:** buckling, bending, appearance defect, textile laminates, fabric wrinkling, polymer foam

## Abstract

Car buyers today are very demanding customers and pay great attention to the quality of purchased goods. They are interested in the quality of the interior as well, for example, the car seats. To avoid customer complaints, the manufacturer needs a sophisticated product testing method that detects any undesirable appearance defects in the textile laminate used for the production of seats in time. The aim of the research was to replace the manual test device used until now with a new methodology for evaluating the appearance defects of textile laminates. The theoretical basis of the methodology for evaluating wrinkled fabric using the well-known buckling theory is described. A newly developed device is introduced. The principle of the new measurement methodology has been patented.

## 1. Introduction

Seat upholsteries are of great importance due to their effect on the driver and passenger comfort and also the aesthetic appeal of the interior [1]. An average of 5–6 m^2^ of fabric (weight 350–450 g/m^2^) are used in cars for upholstery [2]. Most car seat fabric is made from polyester fibre, but there are still relatively small amounts of nylon, wool, and acryl use [3]. There are many studies about the properties of car seat fabrics: the air permeability in car seat covers [4], compression stress-strain [5], the tearing strength and elongation of cover seat fabrics [6], the abrasion resistance [7] and non-flammability of seat covers [8].

Car seat upholstery is generally made of different structures. Seat upholstery fabrics are used as trilaminar structures [9]. One layer of seat upholstery is lamination foam, mostly made of polyurethane (PUR) due to the comfort and protection level given by PURs, which is not achieved by any other single structure. Lamination foams made of PUR are generally obtained by an isocyanate reacting with polyols in the presence of blowing agents. The second part is the face fabric, which is mainly a woven or knitted polyester fabric. Another part of seat upholstery is the backing scrim, which is generally produced from polyester or polyamide [1].

The highest share of fabric in vehicles is upholstery fabric, approximately 85%. Fabrics for upholstering the vehicle interior are mostly of synthetic origin, but it is recommended that covers used for making original covers should be of natural raw materials, which provide a comfortable feeling to the touch [10]. The types of fibres used in the construction of the car seat fabric are shown in articles [2,11], polyester (polyethylene terephthalate), nylon 6.6 (nylon 6 is also used in some countries), and polypropylene fibres. Polyester, discovered in 1941 and commercialized in 1948, is the second largest consumed fibre after cellulosic (natural and regenerated). With polyester being hydrophobic, there is no significant change in its tensile properties when it is wetted, which is a suitable feature for seat covers [12]. Authors [13] investigate the effects of filament cross section on the performance of automotive upholstery fabrics. Changing the cross section of the filaments also affects the yarn structure since it changes the flexural rigidity of the filaments and the drag forces acting on them.

There are many studies about the wrinkle resistance of fabrics in the literature, but little has been written about the wrinkle resistance of fabrics used in seat covers, say the authors of the study. The wrinkle resistances of 10 laminated and non-laminated fabrics were measured at regular intervals in this study. The method of using softening for finishing was found to be the most convenient one to produce seat cover fabrics as highly wrinkle resistant [14].

A foam significantly affects the bending properties of seat covers. When an elastomeric foam is compressed, it first deforms in a linear-elastic way; then, its cells buckle to give nonlinear elasticity; and, finally, the cells collapse completely, and the stress rises rapidly as opposing cell walls are forced together. A plastic foam behaves in a somewhat similar way except that now linear elasticity is followed by plastic collapse and, finally, the forcing together of the cell walls [15,16,17]. Flexible polyurethane foam modelling and the identification of viscoelastic parameters for automotive seating applications are presented in the article by the authors [18]. The influence of structural anisotropy on the mechanical behaviour in the compression of polyurethane foams was studied by means of experiments and computational micromechanics simulations in the articles of authors [19,20,21].

Customers today are very demanding and pay great attention to the quality of purchased goods. This creates pressure on manufacturers who must improve their quality control systems and introduce more sophisticated methods of evaluating the quality of manufactured products. This behaviour between the customer and the manufacturer is most evident, especially in the automotive industry.

This is the reason this new methodology for evaluating the appearance defects of textile laminates was developed, and this new test device is presented here. The aim was to replace the manual test device used until now, see company standard [20].

Textile laminates used for the production of car seat covers are inhomogeneous materials that are prone to shape instability. A laminate is usually composed of three layers: a top fabric, polyurethane foam, and a bottom knitted fabric. Some unsuitable properties of these listed materials can cause shape instability of the final laminate. This results in an appearance defect characterized as wrinkling of the textile in the final product, see Figure 1. To avoid customer complaints, the manufacturer needs a sophisticated product testing method that will detect an undesirable appearance defect in the textile laminate in time.

The investigated appearance defect of a textile laminate is still evaluated according to the Spanish standard [22] by a simple method based on the principle of bending the textile. A 170 × 70 mm^2^ textile sample is placed on a small jig, see Figure 2, consisting of two arms. One arm is stationary, and the other one can rotate. In the starting position, both arms are horizontal; the rotation angle is 0°. Subsequently, one end of the textile sample is fixed to the stationary part of the jig, and its other end remains free. The experimenter starts turning the movable arm of the jig up to the moment when a wrinkle appears on the bent fabric from one edge to the other. At that moment, the experimenter reads the rotation angle on the protractor. According to the standard, the boundary angle for determining the quality of the textile is 50°, see Figure 3. A defective textile, i.e., NOK, shows an angle ≤50°; in contrast, a high-quality textile, i.e., without any defect, shows angle values >50°.

Appearance defect testing according to the current standard shows the following deficiencies:The manual way of operating the device causes:A time-consuming procedure for the experimenter;The introduction of human errors into the measurement, for example, a different speed of the rotating arm movement, an incorrect angle reading.A low number of stress cycles for each sample is used, i.e., only 2–6 rotations of the arm before measuring the angle for the observed fracture on the laminate textile;The quality of the lighting also affects the correct observance of an appearance defect on the textile;The colour and texture of the top fabric of the laminate textile make it harder to exactly assess the defect of the laminate textile.

## 2. Methods

In the first phase of the research, a device based on the existing measurement principle was produced with the purpose of eliminating the above-mentioned drawbacks.

### 2.1. Innovation of a Device

The newly made prototype of the device called SAHF-W, Figure 4, is automated, and all measurements take place in a light box. The fabric is placed on a jig consisting of two arms, a rotating one (1) and a stationary one (2), then it is fixed to the stationary part using clamps (3). The rotating arm is driven by a stepper motor in steps that are controlled by a control unit. At each step, the arm turns by an angle of 2°, and at the same time, a camera (5) records the image of the deformed fabric. The resulting photographs are evaluated by the experimenter. A program code was also written in the Matlab environment to replace the experimenter’s work. Each image was converted from a grayscale image to a binary image, see Figure 5, and the course of the defect occurrence on the textile was analysed using filters.

The image processing method worked for monochromatic woven samples. However, when we started testing samples in which the top fabric had a distinctive large or unsmooth pattern, problems with identifying the defect occurred. The examined defect began to blend with the textile pattern, and the shape of the textile pattern merged with the examined defect, which caused an error in the evaluation, and the defect was not detected, as shown in Figure 5. Therefore, we started looking for a new way of appearance defect evaluation.

### 2.2. Development of a New Method of Evaluating an Appearance Defect

In order to eliminate the influence of the pattern and colour of the fabric on the evaluation of its fracture wrinkling, we decided to develop a different testing method. We started from the well-known theory of bending the textile into a loop and from other ways of stressing the textile during bending, as described in the work by Kopp [23]. Various types of shifting and rotation of the arms, including the fixing of the textile sample, can also be found in the work of Kim et al. [24].

A new device operating on a different principle than that defined by the standard [22] was designed. This device, as can be seen in Figure 6, called SAHF-L, consists of a jig with clamps, a cogged belt, which enables the concentric movement of the clamps, and a stepper motor, which drives the movement of the clamps. The device is again placed in a light box, and the profile of the fabric is photographed.

Both ends of the textile were clamped in the clamps, which then concentrically moved towards each other by a defined distance of 5 mm from each side. The clamps moved from the initial position corresponding to the length of the sample of 170 mm to the final position, which was equal to the value of 20 mm, see Figure 7. The size of the tested fabric was 170 × 70 mm^2^. The shape of the bent fabric profile was photographed, and the resulting geometric shapes, i.e., the height, width, and area, were evaluated. We hypothesized that defective textiles would show significant differences in these parameters. However, the stated hypothesis was not confirmed. Defective textiles did not show significantly different values of geometric parameters from high-quality textiles. It was necessary to look for another way to test and evaluate appearance defects.

We found inspiration in the theory of textile buckling [25], which led us to the following experiment. We clamped a high-quality and defective textile sample side by side in the clamps and concentrically moved the clamps closer to each other, step by step. At a certain point in the movement, the high-quality fabric still held the shape of a smooth arc, but the low-quality fabric collapsed. A new methodology for textile defect measurement was created based on the theory of textile buckling and the loss of shape stability of the textile. The shape of the profile of the high-quality and low-quality textiles can be seen in Figure 13.

#### 2.2.1. Modelling Textile Laminate Buckling

We come across the theory of fabric buckling in many publications: Clapp [25] studies the influence of fabric weight on its buckling; Gao [26] uses buckling theory to calculate the bending properties of the down. A nonlinear bending rigidity model and a study of the critical buckling load can be found in the work of the authors [27].

The authors [28] analyse the buckling of textile under uniaxial stress in the direction of its weft or warp, which is clamped in the universal Tenso Lab Tester. In this paper, the buckling of fabric without holes and with holes was analysed. It was proven that the main modes of deformation in the buckling of fabrics with holes are the fabric bending rigidity, fabric design, and the ratio of the volume of the hole to the volume of the sample.

For the purposes of our research, we present the following theory.

A textile laminate test sample of length L is subjected to horizontal forces F at both ends, as shown in Figure 8a.

When the textile laminate is bent, the textile at the bottom side of the laminate with its length l << L is stressed, this part is magnified in Figure 8b. For most of the investigated laminates, which exhibited an appearance defect, the bending rigidity of the fabric located at the bottom of the laminate is significantly greater than the stiffness of the polyurethane foam.

As can be seen in Figure 8b, this fabric is pressed into the PUR foam mass, i.e., the textile laminate loses its shape stability.

In the first approximation, we investigate the problem of local stability on a one-step fabric model according to Figure 8c.

The bending moment equilibrium equation of the free arm of length l/2 in Figure 8d is:(1)$$M_{0}+M_{1}+\frac{P}{2}.\frac{l}{2}.\cos\varphi -F_{t}.\frac{l}{2}.\sin\varphi =0$$

In the joints of the model, we consider the elastic rotation spring *k* [N.m/rad] and damping with coefficient *b* [N.m.s/rad]; we assume the dependence of the torques on the angle *φ* and $\dot{\varphi }=d\varphi /dt$, respectively.
(1a)$$M_{0}=k.\varphi +b.\dot{\varphi },\;\;\;\;\;\;M_{1}=k.2.\varphi +b.2.\dot{\varphi \;\;}.$$

The foam pressure is proportional to the amplitude of the fabric buckling and the foam rigidity *χ* [N/m] concentrated in the vertical force,
(1b)$$P=\chi .\frac{l}{2}.\sin\varphi \;\;.$$

The horizontal force in the fabric *F_t_* is a fraction of the total horizontal force *F* in the cross section of the textile laminate, Figure 8b, *F_t_* decreases from *F*_0_ to *F*_1_ as a function of the inclination angle of the arm *φ*.
(1c)$$P=\chi .\frac{l}{2}.\sin\varphi \;\;.$$

After substituting (a), (b) into (1), we obtain the equation:$$3.k.\varphi +3.b.\dot{\varphi }+\frac{\chi .\;\;l^{2}}{16}.\sin\left(2\varphi \right)-F_{t}\left(\varphi \right).\frac{l}{2}.\sin\varphi =0\;\;,$$

After simplification, we obtain the equation:(2)$$\dot{\varphi }=\frac{k}{b}\left(\frac{F_{t}\left(\varphi \right).l}{6.k}\sin\varphi -\varphi -\frac{\chi .\;l^{2}}{48.k}\sin2\varphi \right)\;\;.$$

After introducing the critical fabric force (for *b* = 0, *χ* = 0) *F_kr_* = 6. *k*/*l* and designating the dimensionless factor:(2a)$$\dot{\varphi }=\frac{k}{b}\left(\frac{F_{t}\left(\varphi \right).l}{6.k}\sin\varphi -\varphi -\frac{\chi .\;l^{2}}{48.k}\sin2\varphi \right)\;\;.$$

Equation (2) simplifies to:(3)$$\dot{\varphi }=\frac{k}{b}\left(f\left(\varphi \right).\sin\varphi -\varphi -p.\sin(2\varphi \right)\;\;.$$

Where the force applied to the fabric element *F_t_* becomes dimensionless according to (c).
(4)$$f\left(\varphi \right)=f_{0}-\left(f_{0}-f_{1}\right).\frac{\varphi }{\varphi_{max}}\;\;\;\;,$$

As a special case, we obtain a conservative equation from relations (3) and (4) for *b* = 0 and *f*_1_ = *f*_0_, respectively:(5)$$f_{0}=f_{0}\left(\varphi ,p\right)=(\varphi +p.\sin\left(2\varphi \right)).\frac{1}{\sin\varphi }\;\;\;\;.$$

Using the MathCad11 software environment, the plot of the functions *f*_0_(*φ*, *p*) for *p* = 0, 0.3, 0.6 the Equation (5) is shown in Figure 9. The functional dependence without the influence of the polyurethane foam (*p* = 0) is highlighted in black in the plot.

According to Equations (3) and (4), the time progression of the arm inclination angle *φ* in Figure 10a and the dimensionless fabric force *f*(*t*, *φ*) in Figure 10b, respectively, are in the time interval t from 0 to 0.5 s.

#### 2.2.2. Investigation of Fabric Stability Loss—Model 2D

The individual layers of the laminate affect its final properties. It is known from the theory of simple bending that, in a layered material, different values of EJ (bending rigidity) in each layer can cause a displacement of the neutral plane, which affects the resulting bending rigidity of the laminate, Figure 11. The appearance defect that we are examining on the textile is visible when we turn the laminate over so that the knitted fabric becomes the top layer and the woven fabric the bottom layer, see Figure 11a. The different bending rigidity of each layer of the laminate becomes evident when the laminate is bent as a whole. If the top layer (woven fabric) is flexural and soft, this fabric will most likely not have the tendency to create bending fractures and push through the second layer, which is the PUR foam. If the top layer of the textile laminate is significantly stiffer in bending, the value of the neutral plane will shift, and the laminate will begin to show a tendency for the top layer—the fabric—to break into the second layer of PUR foam, as demonstrated in Figure 11b. The bending rigidity of the fabric is affected by several parameters, for example, the type of fibre used, the structure parameters of yarn or fabric, such as the number of twists of the yarn, the density of weft or warp, the type of weave, etc. The problem of compression plate buckling behaviour of rubber and laminate fabrics is examined in the works of the authors [29,30].

Figure 12 shows the simulated progress of textile buckling of the discretized model of the investigated textile laminate sample in the Working Model 2D program environment. Figure 12a shows the fabric in the starting position before the measurement, Figure 12b shows the buckling without the loss of shape stability, and Figure 12c shows the deformation when the fabric lost its shape stability. The graph shows a sudden change in the inclination angle of the monitored red element of the model at a stressed place.

It must be noted that the actual textile behaviour is much more complex than the substantially simplified computational or simulation models presented here.

## 3. Results

The experiments were performed on both types of devices. Measurement methodologies are described in Section 2.1 and Section 2.2. These experiments were necessary to verify the correct setting of the new measurement methodology. For this purpose, we used two sets of textiles whose quality standards had been verified in practice. The first textile set was rated as NOK, and the second set as OK.

Both tested textile sets had the same structure; they were made of the same material. Originally, both sets of these laminate textiles were rated as NOK, but the second set was chemically treated in a way where the stiff top fabric of the laminate was softened, and therefore the textile was free of appearance defects and showed OK values when tested.

The experiments performed on these textiles made it possible to optimize the parameters for determining the evaluation criteria for the new measurement methodology described in Section 2.2. Before we determined the final way of evaluating the photos of the profile of the tested fabric, it was necessary to carry out a number of experiments and check the behaviour of the textile under different settings of the mechanical stress parameters. It was necessary to find the optimum for:The number of cycles for the maximum concentric sliding of the clamps to cause the maximum bending deformation of the textile before the final scanning of the textile profile;The distance by which the clamp jaws will move concentrically in each step;The number of recorded and processed textile profile images, from which the average value of the boundary angle will be subsequently calculated, see Figure 14.

The standard [20], describing the existing measurement method, recommended only two cycles of textile bending: “Rotate the moveable arm of the device with the placed sample to a position of 90° twice in a row and then rotate it at a speed of 10°/s from 0° until a bending (wrinkling) fracture appears on the textile across its entire width of 70 mm”.

Thanks to the automation, we were able to stress the textile with any number of cycles on the SAHF-W device (Section 2.1). We conducted an experiment with five textiles, where the number of cycles was gradually increased, and then the angle at which a bending fracture appeared across the entire width of the textile sample was measured. The textiles were evaluated after 6, 30, 100, 200, and 500 cycles. The following results were recorded: after six cycles, the textile fracture angle occurred at an interval of 60–58°; afterward, the textile was stressed with 30 cycles, and the angle of the textile fracture was at an interval of 45–41°. With further cyclic stressing of up to 100 cycles, we obtained the textile fracture angle within an interval of 41–39°. When stressing the textile for 200 and 500 cycles, we obtained identical results, i.e., the textile fracture angle at an interval of 39–38°. This analysis shows the following: If we stress the textile with many cycles (100–500 cycles), we will narrow the textile fracture angle interval, but we will increase the testing time. However, we are able to obtain sufficiently reliable results about the quality of the textile after only 30 cycles of stressing the textile. Experiments have shown that cyclic stressing of the laminate detects the rheological properties of the textile and thus enables us to obtain a more reliable evaluation of the appearance defect. In the following experiments for both measurement methods, the number of 30 cycles of stress was chosen before reading the textile fracture angle.

For the new SAHF-L device (Section 2.2), we optimized the size of the concentric shift of the device’s clamps and the number of recorded images. Experiments have shown that a concentric movement of the clamps by 10 mm in each step is optimal for textile buckling. The two resulting images are shown in Figure 13. These exact two images were used for the analysis of the profile shapes while looking for the place where the fabric loses its stability, i.e., the arc shape of the textile profile collapses.

The output was photos of the textile profiles, which were subsequently processed in the Matlab program. The created code enabled us to convert the grayscale textile profile image into a bitmap image (black and white pixels). From this image, the lower curve was obtained, which was used for the final evaluation of the defect, i.e., the calculation of the textile fracture angle at the maximum of the curve in the vertical direction or peak, see Figure 13. It should be noted that the textile fracture angle value on this SAHF-L device is not identical to the textile fracture angle value obtained on the SAHF-W device. The evaluation principle is different.

The resulting value of the angle, according to which the quality of the fabric is evaluated, i.e., OK or NOK, is calculated as an average value from the size of the angle obtained from the images in the first and second phase of the concentric shifting of the SAHF-L device clamps. In the case demonstrated here, the resulting angle equals 146.5°. The results of the evaluation of twenty samples of the same type of textile in the direction of the warp and in the direction of the weft are presented in a box diagram and also in a text file, see Figure 14.

The size for the boundary angle of 150°, which defines a textile with a defect or without a defect by the SAHF-L method, was determined empirically. Hundreds of images of the fabric buckling profile were analysed, and the analysis showed that NOK fabrics show an angle smaller than 150°; for example, the tested fabric type GC-O, see Figure 14.

To verify the correctness of our newly proposed methodology for evaluating the appearance defect of textiles, we measured two sets of textile laminates using both test methods (the standardized method described in Section 2.1 and the new method described in Section 2.2). The produced textile demonstrated an appearance defect, i.e., wrinkling. This fabric was divided into two parts, with one part chemically treated to remove wrinkling fractures (and so improve the quality). The first set of forty samples, called CLA, was a high-quality textile, and the second set, called CLB, was a poor-quality (non-treated) textile. With the innovative SAHF-W method (measurement carried out according to the standard), high-quality textiles reach an angle greater than 50°. The high-quality fabric, called CLA, tested in our experiment achieved an average angle of 60°. For the new method measured using the SAHF-L device, a high-quality textile should reach an angle with a value greater than 150°. The average value of the angle calculated for the CLA fabric was 151°.

For the low-quality CLB fabric, we obtained an average angle value of 51° using the SAHF-W method and an average angle value of 143.5° using the SAHF-L method. As can be seen from the results using the SAHF-W method, we were very close to the border of quality determination, but it should be noted that the result can be significantly influenced by the size of the tested sample area. The currently used method (the SAHF-W device), according to the standard, unlike the newly proposed method, evaluates a smaller area of the textile while using the same size of a sample; therefore, there is a high probability that some defective textile samples will not be detected, which also became evident in this case. With the SAHF-W method, the tested part of the sample covers an area of approx. 70 × 20 mm, i.e., the space between the fixed and rotating arm, is approx. 10% of the whole area of the tested sample. However, with the new SAHF-L method, the fabric is fixed to points by clamps, and therefore the unattached tested part of the laminate, on which the defect can occur, covers a significantly larger area, approx. 70 × 130 mm (it is approx. 70% of the whole area of the tested sample), which means there is a greater probability of detecting an appearance defect on the textile.

## 4. Conclusions

A newly designed methodology for measuring an appearance defect of a textile, a defect that causes a bending fracture of the textile on the final product, a car seat cover, was presented. The reasons why a new measurement methodology needed to be designed were described. By implementing this new methodology, the subjective influence of the device operator will be excluded, and the automation of the measurement process will also speed up the testing of textiles and the evaluation of the results. The individual steps of device development and the method of processing the resulting data were described.

The new way of evaluating the appearance of defects was compared with the existing way of evaluating these defects. The new measurement method covers a larger area of the tested sample (it covers 60% more of the tested area) and thus can give more reliable results about the quality of the textile with a higher degree of probability. The measurement principle has been patented [31], and the next goal is to introduce this new measurement method into production.

## Figures and Tables

**Figure 1 polymers-14-04527-f001:**
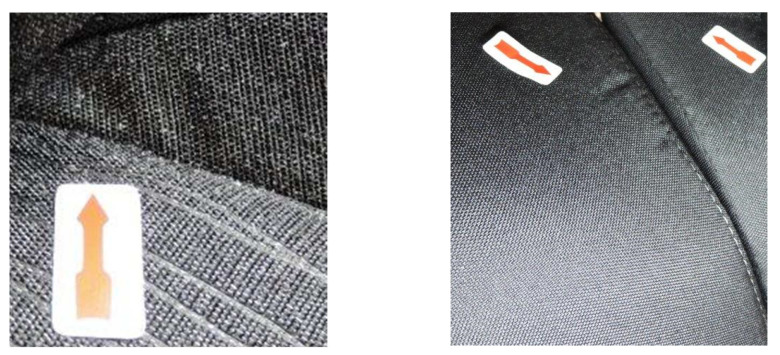
The investigated appearance defect of a laminate textile—fabric wrinkling.

**Figure 2 polymers-14-04527-f002:**
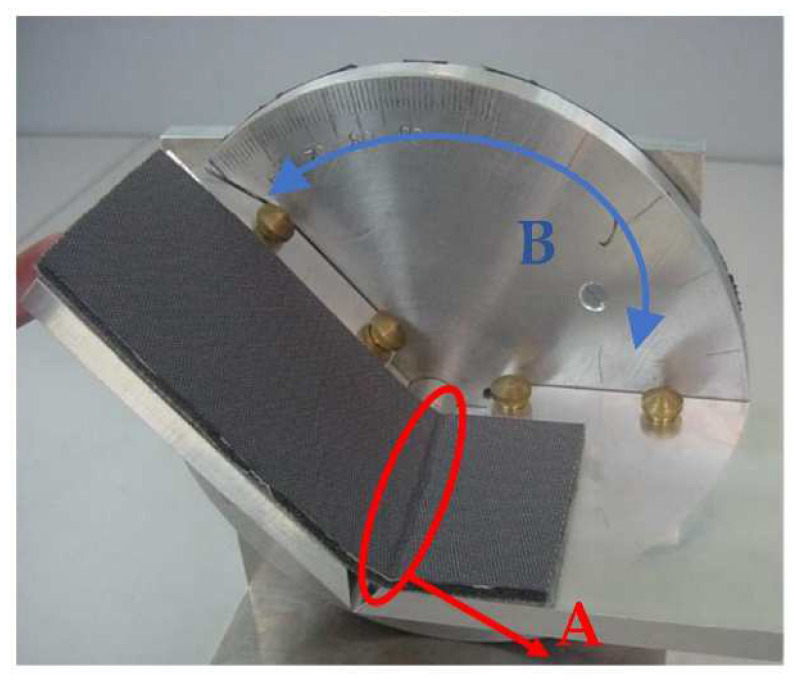
The evaluation of an appearance defect of a textile laminate: (A) formation of wrinkles on the fabric; (B) the moving of the rotating arm with the fabric.

**Figure 3 polymers-14-04527-f003:**
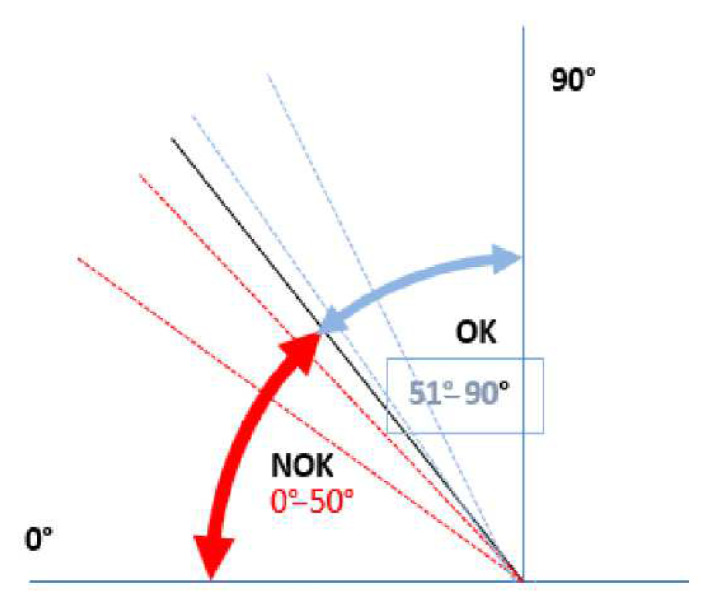
Evaluation criteria for textile laminate defects according to the standard.

**Figure 4 polymers-14-04527-f004:**
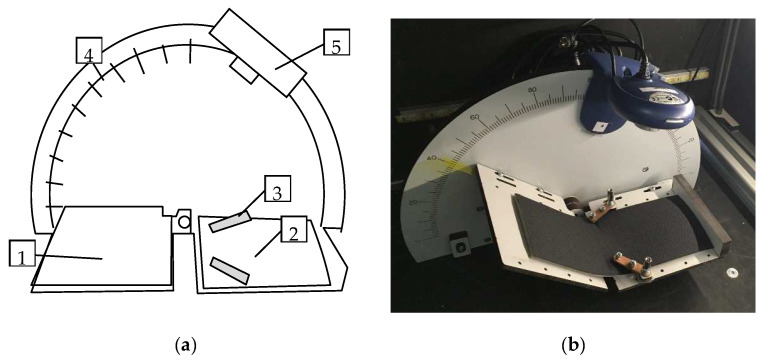
The innovated device called SAHF-W: (**a**) the scheme of device 1—rotating arm; 2—stationary arm; 3—fixed clamps; 4—protractor scale; 5—camera; (**b**) photo of the produced device.

**Figure 5 polymers-14-04527-f005:**
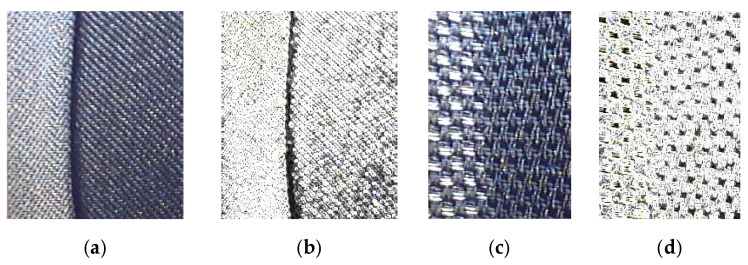
Textile with a defect evaluated using the image analysis method: (**a**,**b**) fabric with a smooth pattern; (**c**,**d**) fabric with an unsmooth pattern.

**Figure 6 polymers-14-04527-f006:**
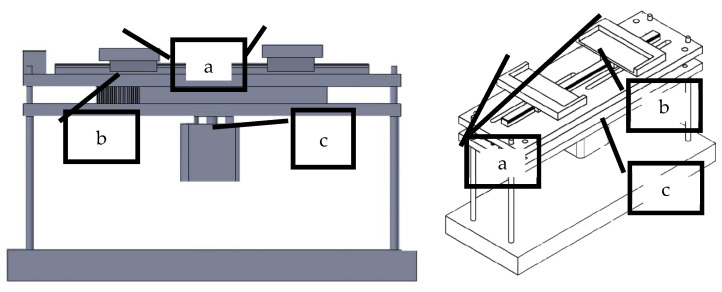
The mechanical device called SAHF-L: (a) sliding clamps; (b) cogged belt; (c) stepper motor.

**Figure 7 polymers-14-04527-f007:**
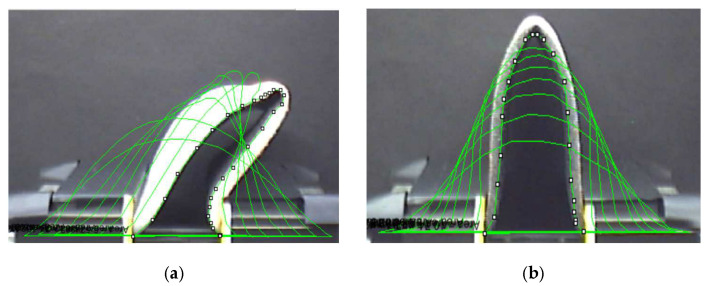
The shape of the bent fabric profile: (**a**) fabric with a defect; (**b**) high-quality fabric.

**Figure 8 polymers-14-04527-f008:**
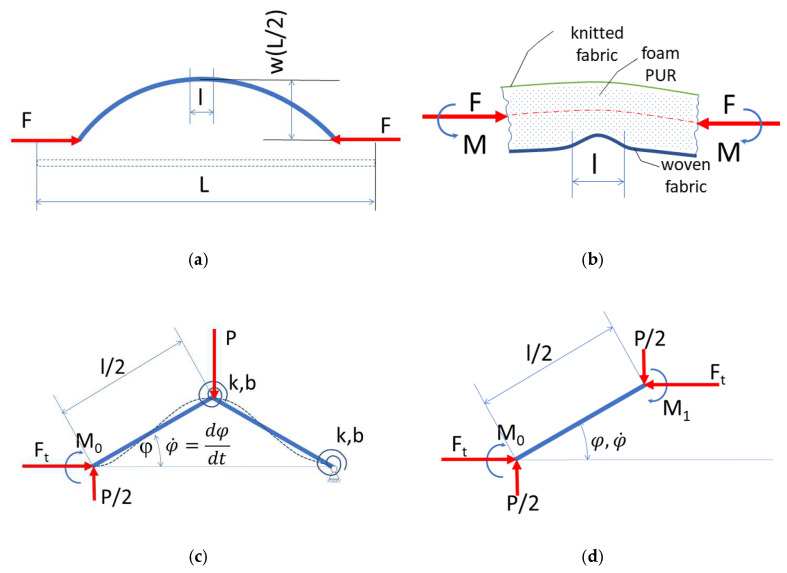
A model of textile laminate test sample of length L: (**a**) a sample of length L is subjected to horizontal forces F at both ends; (**b**) the fabric is pressed into the PUR foam mass, i.e., the textile laminate loses its shape stability; (**c**) a one-step fabric model; (**d**) the bending moment equilibrium equation of the free arm.

**Figure 9 polymers-14-04527-f009:**
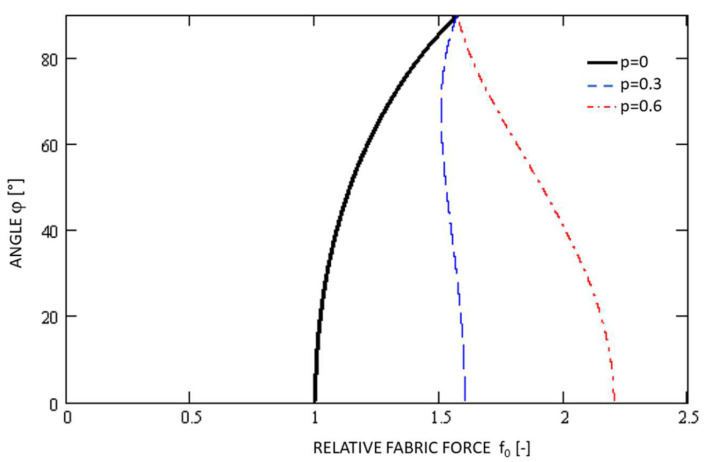
The plot of the functions *φ* = (*f**_0_***, *p*).

**Figure 10 polymers-14-04527-f010:**
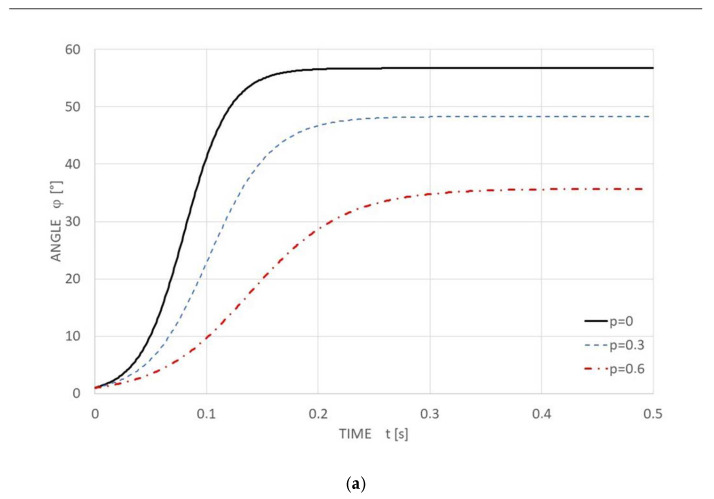
The functional dependency: (**a**) the time progression of the arm inclination angle *φ*.[°]; (**b**) the dimensionless fabric force *f*(*t*, *φ*).

**Figure 11 polymers-14-04527-f011:**
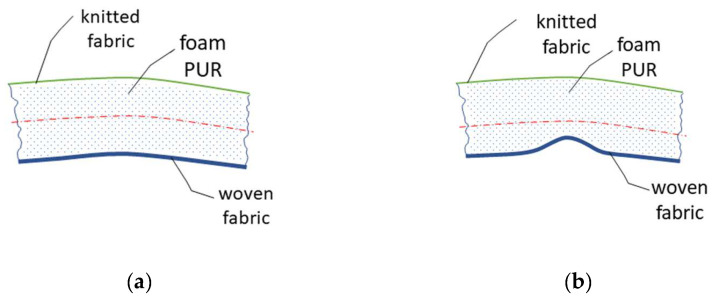
Bending properties of the laminate layer: (**a**) good quality textile; (**b**) textile with a defect.

**Figure 12 polymers-14-04527-f012:**
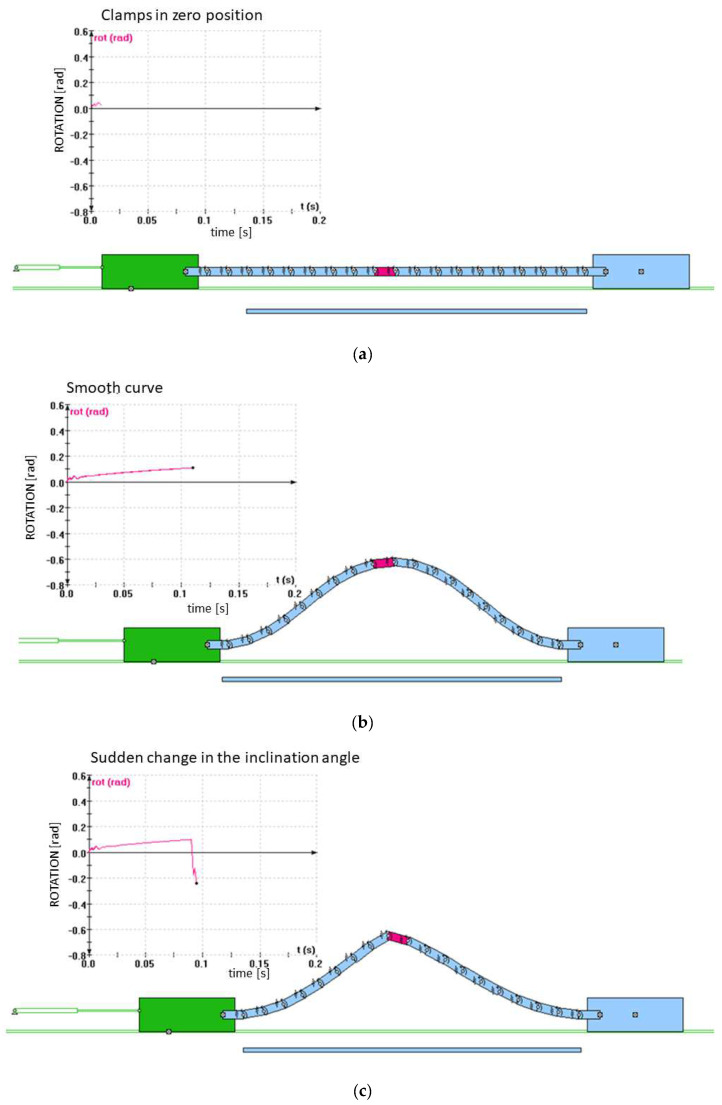
The simulated progress of textile buckling of the discretized model of the investigated textile laminate sample in the Working Model 2D program environment at a clamp distance of 110 mm: (**a**) the starting position; (**b**) the buckling without the loss of shape stability; (**c**) the loss of shape stability.

**Figure 13 polymers-14-04527-f013:**
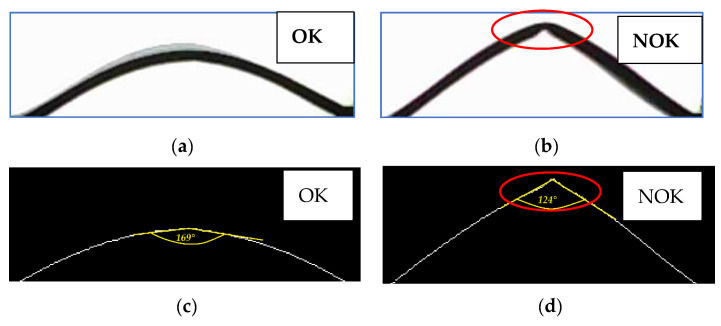
Textile profile photo processing for fabric type GC: (**a**) textile GC-u is OK—without loss of stability; (**b**) Textile GC-o is NOK– loss of stability detected; (**c**) the curve and the deformation angle reading; (**d**) the angle of deformation of the textile after its loss of stability.

**Figure 14 polymers-14-04527-f014:**
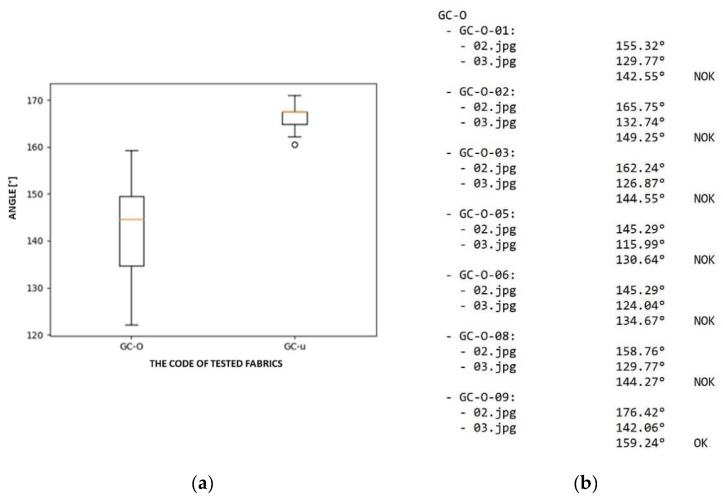
The example of results of the examined textiles: (**a**) the box diagram of results; (**b**) measurement results saved as a text file for the tested fabric type GC-O.

## Data Availability

Partial research reports and a patent. https://patentimages.storage.googleapis.com/fa/a0/d8/2b68c28721c011/CZ308682B6.pdf (accessed on 18 September 2022).

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
