# Peer review of "Investigation of Loss of Shape Stability in Textile Laminates Using the Buckling Method"

_polymers, 2022, doi:10.3390/polym14214527_

Round 1

Reviewer 1 Report

In this manuscript, the author investigated the loss of shape stability in textile laminates using the buckling method. The paper fits within the scope of the journal - polymers; however, some issues should be addressed before a decision is made.

  1. The justification of the study should be presented in the introduction section rather than the methods section. More information about the testing of fabric defect should be presented. The current version mainly focused on foams which are not the main material of interest.
  2. The images in Figure 4 and 14 should be labelled (a) and (b). The appropriate caption for the image on the right hand side should be added.
  3. What is the size of fabric used in the SAHF-L device? It is unclear how the results obtained using a smaller sample is scaled up for the entire car seat upholstery.
  4.  The modes of deformation listed in lines 155 - 157 are not correct. They appear to be the factors influencing the buckling of fabrics.
  5. The equation numbering should be sequential without alphabets.
  6. Line 188: "p" is not dimensionless.
  7. The axis labels in Figure 9 and 10 are unclear. Legends should not be placed on the axis.
  8. How does imperfect bonding and delamination in the laminate affect the accuracy of the presented buckling analysis?
  9. The plot titles in figure 12 should be updated to English language. Axis labels and plot title are also missing in figure 14.
  10. "chapter 2.1 and 2.5" on lines 239, 269 and 286 should be updated to sections or subsections.
  11. The relationship between the angle presented in section 2.4 and those presented in figure 14 is unclear. GC-O and GC-u were not defined. More details needed to understand the data presented in this figure.
  12. Please clarify how the size for the boundary angle of 150° was determined empirically.

Author Response

  1. The justification of the study should be presented in the introduction section rather than the methods section. More information about the testing of fabric defect should be presented. The current version mainly focused on foams which are not the main material of interest.
    a) The justification of the study was moved to the introduction section
    b) Several studies on fabric testing have been added to the introduction section
  2. The images in Figure 4 and 14 should be labelled (a) and (b). The appropriate caption for the image on the right hand side should be added.
    Corrected
  3. What is the size of fabric used in the SAHF-L device? Corrected Line 146. It is unclear how the results obtained using a smaller sample is scaled up for the entire car seat upholstery.
    The size of the tested sample is 170x70 mm2. Results of tested samples is applied by statistical analysis for the entire car seat upholstery.
  4.  The modes of deformation listed in lines 155 - 157 are not correct. They appear to be the factors influencing the buckling of fabrics.
    ?? in lines 155 – 157 It was cited from the article.
  5. The equation numbering should be sequential without alphabets.
    Corrected
  6. Line 188: "p" is not dimensionless.
    dimension of "p"  [N/m * m2*rad/N*m];    p[rad]
  7. The axis labels in Figure 9 and 10 are unclear. Legends should not be placed on the axis.
    Figures 9-10 were corrected
  8. How does imperfect bonding and delamination in the laminate affect the accuracy of the presented buckling analysis?
    The number of lamination points and the thickness of the adhesive film will affect the resulting buckling value. More points as well as more thickness of the adhesive film means higher stiffness of the textile laminate.
  9. The plot titles in figure 12 should be updated to English language. Corrected Axis labels and plot title are also missing in figure 14. Corrected

  10. "chapter 2.1 and 2.5" on lines 239, 269 and 286 should be updated to sections or subsections.
    Corrected
  11. The relationship between the angle presented in section 2.4 and those presented in figure 14 is unclear. GC-O and GC-u were not defined. More details needed to understand the data presented in this figure.

The angle is investigated in the region of the maximum of the curve

12. Please clarify how the size for the boundary angle of 150° was determined empirically.

17 types of textiles were tested. one set contained 5-10 warp and weft samples, that's a total of 310 samples. The calculated angles at the peaks of the curve were manual compared with the profile image of the fabric. Loss of fabric shape stability corresponded to an angle value smaller than 150° in 93% of cases (186 out of 200),

Reviewer 2 Report

General Comment: 

An interesting work for narrow specialists that studies the loss of shape stability in textile laminate using the buckling method. The paper can be considered for a potential publication; however, appropriate revisions are needed before publication:

1.      Figures are very good quality but sentences by czech language in Fig. 12 must be deleted.

2.      In my opinion description of results in Figure 9 and Figure 10 must be done.

3.      What is a minimal distance between sliding clamps in device SAHF_L during experiment?

4.      Did you study influence of minimal distance in dependence of material quality?

5.      The Conclusions is interesting, but it presents general formulas. In the conclusions it would be advisable to indicate more numerical results.

Author Response

however, appropriate revisions are needed before publication:

  1. Figures are very good quality but sentences by czech language in Fig. 12 must be deleted.
    Corrected
  2. In my opinion description of results in Figure 9 and Figure 10 must be done.
    Corrected
  3. What is a minimal distance between sliding clamps in device SAHF_L during experiment?

The first phase, the clamps move to each other at a distance of 130mm. Clamps are 20mm apart. This stressing of the fabric is done in 30 cycles. In the second phase, the clamps are moved by 10mm, followed by photographing the fabric profile. and another clamp displacement of 10 mm and the profile of the bulging fabric is photographed.

  1. Did you study influence of minimal distance in dependence of material quality?

Yes, 17 types of textiles were tested. one set contained 5-10 warp and weft samples, that's a total of 310 samples. We searched for the optimal distance to detect the loss of shape stability. The result is a shift of clamps of 10 or 20 mm, respectively

  1. The Conclusions is interesting, but it presents general formulas. In the conclusions it would be advisable to indicate more numerical results.
    Yes.

Round 2

Reviewer 1 Report

The manuscript has been revised in line with my comments and suggestions. It can be accepted for publication after the reference to "chapter" is updated to "section or subsection".

Author Response

It can be accepted for publication after the reference to "chapter" is updated to "section or subsection"

Chapters 2.3 and 2.4 was updated to chapter 2.2.1 and 2.2.2
